# Effects of Steaming on Sweet Potato Soluble Dietary Fiber: Content, Structure, and *Lactobacillus* Proliferation *In Vitro*

**DOI:** 10.3390/foods12081620

**Published:** 2023-04-12

**Authors:** Zhiguo Zhang, Buyu Liu, Xingquan Liu, Weiwei Hu, Chengcheng Zhang, Yang Guo, Weicheng Wu

**Affiliations:** 1Food Science Institute, Zhejiang Academy of Agricultural Sciences, Hangzhou 310021, China; zhangkii@126.com (Z.Z.);; 2College of Food and Health, Zhejiang Agriculture and Forestry University, Hangzhou 311300, China

**Keywords:** steamed sweet potato, soluble dietary fiber, composition, *Lactobacillus* fermentation, short chain fatty acid

## Abstract

The influence of steaming treatment on the soluble dietary fiber (SDF) of sweet potato was investigated. The SDF content increased from 2.21 to 4.04 g/100 g (in dry basis) during 20 min of steaming. The microcosmic morphology of the fractured cell wall indicated the release of SDF components during steaming. The SDF from fresh (SDF-F) and 20 min steamed (SDF-S) sweet potato was characterized. The neutral carbohydrates and uronic acid levels in SDF-S were significantly higher than SDF-F (59.31% versus 46.83%, and 25.36% versus 9.60%, respectively) (*p* < 0.05). The molecular weight of SDF-S was smaller than SDF-F (5.32 kDa versus 28.79 kDa). The probiotic property was evaluated by four *Lactobacillus* spp. fermentation in vitro with these SDF as carbon source, using inulin as the references. SDF-F showed the best proliferation effects on the four *Lactobacillus* spp. in terms of the OD_600_ and pH in cultures, and the highest production of propanoic acid and butyric acid after 24 h fermentation. SDF-S presented higher *Lactobacillus* proliferation effects, but slight lower propanoic acid and butyric acid production than inulin. It was concluded that 20 min of steaming released SDF with inferior probiotic properties, which might derive from the degraded pectin, cell wall components, and resistant dextrin.

## 1. Introduction

Sweet potato (*Ipomoea batatas* [L.] Lam.) is one of the most consumed crops globally and plays an important role in fulfilling the continuously increasing food demand of the world population [1]. As the main edible part, its pulp is a good dietary source of carbohydrates and numerous phytochemicals, including protein, amino acids, polyphenols, polysaccharides, anthocyanins, and β-carotene [2]. These compounds are beneficial to health and are highly desirable in the human diet. It is generally regarded as a healthy food for gut disorders such as constipation, diarrhea, and inflammatory bowel disease, due to its high content of dietary fiber (DF) [2,3,4].

DF has been recognized as indigestible polysaccharides with the degree of polymerization (DP) > 10 [5], such as β-glucan, inulin, pectin, resistant dextrin, cellulose, hemicellulose, etc. DF is usually classified by its solubility in water into insoluble dietary fiber (IDF) and soluble dietary fiber (SDF) [6]. In recent decades, the beneficial effects of DF intake on the gastrointestinal microecological homeostasis have become a topic of interest to consumers, and SDF is superior to IDF in gastrointestinal function regulation [7,8]. Currently, a significant focus of research is projected on maintaining the balance of intestinal flora and promoting the growth of beneficial microorganisms, e.g., *Lactobacillus*, one of the most abundant beneficial microorganisms in the human intestinal tract [9,10]. The species are closely associated with human gastrointestinal health, including *L. plantarum*, *L. reuteri*, *L. acidophilus*, *L. rhamnosus*, *L. casei*, *L. salivarius*, *L. ingluviei*, etc. [11,12,13]. *Lactobacillus* spp. can utilize DF from daily dietary intake and produce a series of beneficial metabolites. Short chain fatty acids (SCFA) are the main beneficial products of *Lactobacillus* spp. fermentation, such as propionic acid, butyric acid, and valeric acid. Previous literature has demonstrated that sweet potato SDF holds benefits for increasing the abundance of probiotic bacteria (e.g., *Lactobacillus*, *Faecalibaculum*, and *Bifidobacterium*) in vitro and in vivo, as well as improving the SCFA profile in rat intestine by enhancing the percentages of propionate and butyrate [14,15]. SCFA provide energy for the host metabolism and inhibits the growth of pathogenic microorganisms by reducing the gastrointestinal pH, thereby helping to maintain the gastrointestinal microecological homeostasis and treat diabetes, obesity, cancer, inflammation, and immunodeficiency [16,17,18].

As a starchy food, sweet potato is usually cooked in different ways prior to consumption, according to recipes and culinary traditions of the various countries. Available evidence suggests that thermal processing could affect the content, composition, structure, and bioactivities of DF. Lu et al. [19] reported that the thermal treatment on black garlic degraded its polysaccharide molecule weight (Mw) from more than 4 kDa to less than 1 kDa. Dhingra et al. [20] reported that DF content increased in potato after domestic cooking due to the formation of complexes between polysaccharides and proteins in the food or resistant starch in cooked potatoes. Fang et al. [21] reported that steaming and boiling treatment can increase SDF content in highland barley, buckwheat, proso millet, quinoa, sorghum, coix seed, and oat, but decrease it in barley and foxtial millet. Additionally, during fermentation in vitro, oat, quinoa, highland barley, and buckwheat after boiling treatment can produce more SCFA than steaming treatment, while barley, foxtail millet and coix seed showed opposite results. Sweet potato is a desirable source of DF intake, while little was known about the impact of thermal processing on the structure and bioactive property of its DF.

The present study aimed to conduct a comprehensive investigation on changes of the composition, structure, and proliferation effects on *Lactobacillus* spp. during steaming processing, the most common culinary processing method. Sweet potato cultivar “Xinxiang” is a yellow flesh cultivar with thin red skin, which is bred by Institute of Crops and Nuclear Technology Utilization at Zhejiang Academy of Agricultural Sciences. It is very popular for table use in local consumer markets in Southeast China due to its delicate sweet flavor and appetizing color after culinary processing. The SDF contents in sweet potato steamed for different time were measured, then an appropriate steaming time was determined for higher SDF content. Compared to fresh sweet potato, the structural characteristics (chemical composition, molecular weight (Mw), and Fourier transform–infrared (FT-IR) spectrometry features) of steamed sweet potato SDF, as well as the proliferation effects on *Lactobacillus* spp., were systematically evaluated. The morphological features of sweet potato during steaming were also examined using scanning electron microscopy (SEM).

## 2. Materials and Methods

### 2.1. Materials

Sweet potatoes (cultivar “Xinxiang”) were purchased from a local market in Hangzhou, Zhejiang Province, China. Taka-diastase, amyloglucosidase, pancreatin, d-glucose, and inulin were purchased from Yuanye Bio-Tech Co., Ltd. (Shanghai, China). de Man, Rogosa and Sharpe (MRS) broth was purchased from Hope Bio-Technology Co., Ltd. (Qingdao, China). Monosaccharide standards (l-arabinose, l-fucose, l-rhamnose, d-galactose, d-glucose, d-mannose, d-ribose, d-glucuronic acid, and d-galacturonic acid) and SCFA standards (acetic acid, propionic acid, and n-butyric acid) were purchased from Sigma-Aldrich Chemical Co. (Shanghai, China). All other chemicals and reagents used in this study were of analytical grade. *L. plantarum* (RBHZ 68), *L. reuteri* (MFHZ 18), *L. acidophilus* (GFHZ 41), and *L. rhamnosus* (MFHZ 77) were gifted by the Vegetable Processing Laboratory at the Food Science Institute at the Zhejiang Academy of Agricultural Sciences.

### 2.2. Cooking Treatment

Sweet potatoes were washed, peeled, and cut into slices of 1.5 cm in thickness. The slices were steamed for 30 min under atmospheric pressure using a domestic steamer and cooled to room temperature (25 °C). After steaming for 0, 5, 10, 15, 20, 25, and 30 min, the same number of slices were taken out, respectively, and were freeze-dried for SDF determination and morphological observation.

### 2.3. SDF Determination

The contents of SDF in samples were assayed by the enzymatic-gravimetric method following the AOAC 991.43 [22].

### 2.4. Morphological Observation

The microcosmic morphology was recorded using an SEM analyzer (Regulus 8100; Hitachi Ltd., Tokyo, Japan) at an accelerating voltage of 3 kV [23]. The samples were placed on a metal stub with double-sided adhesive tape and sputter-coated with gold in a vacuum sputter coater to make the samples conductive. Representative images were taken at 200×, 500×, and 1000× magnification.

### 2.5. Preparation of Sweet Potato SDF

Based on the results of starch digestibility and DF contents, an appropriate steaming time (20 min) was determined, and the SDF was extracted for investigating its alteration in composition, structure, and proliferation effects on *Lactobacillus* spp. SDF was prepared using ultrasonic-assisted extraction, and the parameters were selected according to our previous study. Freeze-dried sweet potato was grounded into powder, weighed, and mixed with distilled water in a 1:15 ratio (*w*/*v*). Extraction was conducted in an ultrasonic bath processor (Shumei KQ-400DB, Kunshan, China) at 400 W for 40 min. The temperature was set at 40 °C, but it rose from 40 °C to 51 °C during the 40 min of extraction due to the thermic effect of the ultrasonic equipment. Complex enzymatic hydrolysis was used to remove starch and protein components. The sample was first hydrolyzed with 2% (*w*/*w*) Taka-diastase at 60 °C for 1.5 h with constant stirring. Then, the pH of the sample solution was adjusted to 4.5, and 1% (*w/w*) amyloglucosidase was added to the sample to remove the residual starch, with constant stirring at 60 °C for 1 h. Afterward, the pH of the sample solution was adjusted to 7.0, and 0.5% (*w*/*w*) pancreatin was added to the sample for hydrolysis, with constant stirring at 40 °C for another 1 h. The sample solution was centrifuged at 4000 rpm (3500× *g*, Cence H2100R, Changsha, China) for 30 min at room temperature. The supernatant was collected, concentrated, and precipitated with 95% ethanol overnight. Finally, the precipitate was collected by centrifugation (4000 rpm/3500× *g*, 4 °C, 10 min) and freeze-dried. The SDF collected from steamed sweet potato was denoted as SDF-S, and the SDF from fresh sweet potato was denoted as SDF-F.

### 2.6. Characterization of Sweet Potato SDF

#### 2.6.1. Chemical Components Analysis

The neutral carbohydrate content in sweet potato SDF was measured by the phenol-sulfuric acid method, using d-glucose as the standard [24]. The uronic acid content in sweet potato SDF was measured with the m-hydroxybiphenyl method, using d-galacturonic acid as the standard [25]. The total carbohydrate content of SDF was calculated as the summation of neutral carbohydrate content and uronic acid content, as proposed by Hu et al. [26]. Iodine reaction was performed to detect the residual starch or its hydrolysate in sweet potato SDF. The SDF solutions were compared with potato starch solution to identify the existence of starch [27].

#### 2.6.2. Mw Determination

The Mw and homogeneity of SDF were determined via size exclusion chromatography coupled with multi-angle laser light scattering (SEC-MALLS, λ = 658 nm, Wyatt Technology Corporation, Santa Barbara, CA, USA). A TSKgel G5000PWXL column (30 cm × 7.8 mm ID, 10 µm; Tosoh Corporation, Tokyo, Japan) and a TSKgel G3000PWXL column (30 cm × 7.8 mm ID, 6 µm; Tosoh Corporation, Tokyo, Japan) were employed, along with a refractive index detector. The sample was prepared by ultra-pure water and filtered through a 0.22 μm filter before injection. The chromatographic conditions were as follows: column temperature, 25 °C; mobile phase, 0.1 mol/L NaNO_3_; flow rate, 0.5 mL/min; and injection volume, 100 μL. The data were analyzed using Astra 7.3.2 software package, and the refractive index increment (dn/dc) value was determined to be 0.138 mL/g, in accordance with the literature [28].

#### 2.6.3. Monosaccharide Composition Analysis

Monosaccharide composition of SDF was determined by high-performance liquid chromatography (HPLC) [29]. The samples were hydrolyzed by 4 mol/L TFA at 110 °C for 5 h in the sealed test tube. After incubation, the tubes were cooled, and each reaction mixture was dried by nitrogen. The dried hydrolyzed samples and standard substances were added 0.05 mL of 0.5 mol/L methanolic 1-pheny-3-methyl-5-pyrazolone solution and 0.05 mL of 0.3 mol/L NaOH; then, the samples were incubated at 70 °C for 1 h. Standard substances including rhamnose, arabinose, mannose, ribose, glucose, galactose, fucose, glucuronic acid, and galacturonic acid were treated identically to the samples. After that, mixture was neutralized with 0.05 mL of 0.3 mol/L HCl and added 1.5 mL trichloromethane. The samples were mixed thoroughly and kept standing for 20 min. The extraction process was repeated in triple and carefully removed the organic phase each time. Ten microliter of the resulting aqueous phase was analyzed on an Agilent C18 column (4.6 mm × 250 mm × 5 μm; Agilent Technologies, Inc., Santa Clara, CA, USA) connected to a HPLC system (1200 Series; Agilent Technologies, Inc., Santa Clara, CA, USA). The column mobile phase was consisted with 82% 0.1 mol/L phosphate buffer (pH 6.8; solution A) and 18% acetonitrile (solution B). The wavelength for UV detection was 254 nm and the column temperature was held at 25 °C.

#### 2.6.4. FT-IR Analysis

An FT-IR spectrophotometer (FTIR-8900; Shimadzu Corporation, Kyoto, Japan) was used to analyze the chemical bonds and functional groups of sweet potato SDF. Two milligrams of dried sample were ground with 100 mg KBr into a pellet and scanned in a range of 4000–400 cm^−1^ [30].

### 2.7. Proliferation Potential of Sweet Potato SDF on Lactobacillus spp.

#### 2.7.1. Growth Stimulation

The frozen strains were activated by growing them in sterilized MRS broth. Into the broth, 4% (*v*/*v*) of inoculum was inoculated and incubated at 37 °C for 24 h. The strains were sub-cultured twice according to the same procedure until the optical density (OD_600_) of the cell suspension was approximately 1.5.

The non-carbohydrate MRS broth was prepared by the formula reported by de Man et al. [31], with modifications to compare the proliferation effects of different carbon sources. It contained 10.0 g/L peptone, 5.0 g/L beef powder, 5.0 g/L yeast extract, 2.0 g/L ammonium citrate, 5.0 g/L sodium acetate, 2.6 g/L dipotassium hydrogen phosphate trihydrate, 0.05 g/L manganese sulfate, 0.2 g/L magnesium sulfate, and 1.0 mL/L of Tween 80. SDF-S, SDF-F, inulin (positive control), and glucose (negative control) were spread in a thin layer onto Petri dishes and sterilized by ultraviolet light in a biosafety cabinet for 1 h (30 min on each side). The samples were dissolved in water (5%, *w*/*v*), and added into sterilized non-carbohydrate MRS broth to a final concentration of 1% (*w*/*v*). The activated strains were cultured and incubated in the non-carbohydrate MRS broth, and the growth of each strain was monitored by measuring the OD_600_ and pH of the culture media every 2 h during incubation.

The MRS broth of four strains with different carbon sources were collected after 24 h of fermentation and centrifuged at 12,000 rpm, 4 °C, for 10 min to obtain the supernatant. Carbon source utilization ratio of four strains with different carbon sources were determined by the total carbohydrate content in supernatant via the method mentioned in Section 2.6.1.

#### 2.7.2. SCFA Profile

The SCFA, including acetic acid, propionic acid, and butyric acid were measured by gas chromatography (GC) [32]. Samples of the four strains in MRS broth with different carbon sources were collected after 24 h of fermentation and centrifuged at 12,000 rpm for 10 min. The supernatant was collected for GC analysis. To prepare a crotonic acid metaphosphoric acid solution, 0.6464 g of crotonic acid was added into 100 mL of 2.5% (*w*/*v*) metaphosphoric acid solution, and 500 μL of supernatant was combined with 100 μL crotonic acid metaphosphoric acid solution and acidified at −20 °C for 24 h. The acidified supernatant was centrifuged at 12,000 rpm for 5 min, and the final supernatant was filtered and used for SCFA concentration detection. Separation was carried out on a GC-2010 plus GC system (Shimadzu Corporation, Kyoto, Japan) with a DB-FFAP column (Agilent Technologies, Inc., Santa Clara, CA, USA) and a hydrogen (H_2_) flame ionization detector. The injection volume was 1.0 μL and nitrogen (N_2_) was used as the gas carrier at a flow rate of 12.0 mL/min. The flow rates of air, H_2_, and N_2_ in detector were 400.0, 40.0, and 30.0 mL/min, respectively. The temperature of the injector and detector was kept at 250 °C. The oven temperature program was as follows: initial column temperature of 70 °C, increased to 180 °C at 15 °C/min, and increased to 240 °C at a rate of 40 °C/min.

### 2.8. Statistical Analysis

For all the experiments, measurements were performed in triplicate. Data collected from the experiments were firstly subjected to the Shapiro–Wilk test to examine the normality. The data with normal distribution were expressed as mean ± standard deviation (SD), and the difference among groups were determined using one-way analysis of variance (ANOVA) by Duncan test. The data with non-normal distribution were expressed as medium ± SD, and the Kruskal–Wallis test was conducted to detect the differences among groups by comparing the data distribution. All the analysis were conducted by using IBM SPSS program version 23.0 (IBM SPSS Inc., Chicago, IL, USA). Figures were generated with OriginPro 2017 (OriginLab Co., Northampton, MA, USA).

## 3. Results and Discussion

### 3.1. SDF Content

SDF was extracted from sweet potato with an ultrasonic assisted method, and the yield of SDF in dry basis presented as the SDF content. The SDF content in fresh sweet potato (at 0 min) was determined to be 2.21 ± 0.40 g/100 g. Changes of the SDF content in sweet potato during the 30 min of hydrothermal treatment are shown in Figure 1. There was minor increasing in the SDF content during the first 10 min. While it increased significantly from 2.88 ± 0.26 to 4.05 ± 0.33 g/100 g during the 10–15 min of steaming (*p* < 0.05) and kept almost stable during the 15–20 min of steaming. The subsequent steaming from 20 to 30 min led to a decrease from 4.05 ± 0.49 to 3.69 ± 0.38 g/100 g, though the changes were not significant.

The increase in the SDF content has been observed in boiled oats and barley, cooked beans, and microwaved whole grain oats [21,33,34]. Dhingra et al. reported that the increase in dietary fiber may be because of the formation of complexes between polysaccharides and proteins in the food or resistant starch in cooked starchy materials [20]. Garcia-Amezquita et al. reported that thermal processing might cause the degradation of carbohydrate polymers into soluble small molecules [35]. Considering the possible insoluble nature of polysaccharide-protein complex, the significant rising of the SDF content during 10–15 min of steaming might be the result of the accumulated hydrothermal effect on the breakdown of the insoluble carbohydrate polymers. However, the decreasing of SDF content during the 20–30 min of steaming might be due to the further degradation of SDF into smaller fragments, which could not be sedimented in 70% ethanol solution [33].

### 3.2. Morphological Characteristics

The morphological images of fresh and steamed sweet potato cells were recorded in Figure 2. The original cell structure of the fresh sweet potato (0 min) (Figure 2A) and the sweet potato steamed for 5 min (Figure 2B) was largely maintained. Numerous starch granules remained in cells, with irregular shape and small depressions or pores on the surface. After 10 min of steaming (Figure 2C), the sweet potato cells were completely filled with expanded starch granules, which formed an amylose and amylopectin reticulum [36]. At that moment, the cell structure was still observable. After 15 and 20 min (Figure 2D,E), the cell structure was apparently fractured due to the compression of expanded starch. The granule structure of starch was disintegrated, and microfibrillar-like structures were observed on the surface of starch, which might suggest the formation of SDF. After 25 and 30 min (Figure 2F,G), the sample had more microfibrillar-like structures, indicating the subsequential degradation of SDF.

### 3.3. Characterization of Sweet Potato SDF

#### 3.3.1. Chemical Components, Mw, and Monosaccharide Composition

The SDF-F and SDF-S were extracted from the fresh and sweet potatoes steamed for 20 min, respectively. The negative results of the iodine reaction means that no starch component was detected in the two SDF samples (Figure 3). The chemical composition of the two SDFs is shown in Table 1. The total carbohydrate content in the SDF-S was significantly higher than that in the SDF-F (84.68 g/100 g versus 56.43 g/100 g) (*p* < 0.05). The concentration of neutral carbohydrate was significantly higher in the SDF-S than that in the SDF-F (59.31 g/100 g versus 46.83 g/100 g) (*p* < 0.05), in addition to the concentration of uronic acid (25.36 g/100 g versus 9.60 g/100 g) (*p* < 0.05). The Mw of the SDF-F and SDF-S was 28.79 kDa and 5.32 kDa, respectively, and the polydispersity indices (Mw/Mn, Mn is the number averaged Mw) of the SDF-F and SDF-S were 1.35 and 1.61, respectively. Associated with the morphological changes of the cell structure during steaming (Figure 2), the decreased Mw confirmed the degradation of the SDF.

Nine monosaccharides were detected in the SDF-S, including glucose, galactose, arabinose, rhamnose, mannose, ribose, fucose, glucuronic acid, and galacturonic acid, while no fucose could be detected in the SDF-F. Both SDF-S and SDF-F were mainly composed of glucose, which accounts for over ninety percent of their total monosaccharides. The significantly higher extraction yield of the SDF-S than the SDF-F (Figure 1) meant that a large amount of neutral carbohydrate was released during steaming. Similar phenomena have been documented by Dong et al. in steamed whole grain oats [34]. The possible sources of the released neutral carbohydrate were the soluble β-glucan released from the cell wall or the enzyme-resistant dextrin in raw material [37,38]. Resistant dextrin has been identified in other starchy tube crops, such as tapioca and Chinese yam, showing the features of high glucose concentration and low molecular weight [39,40]. In accordance with the uronic acid content found by colorimetric determination, the galacturonic content in SDF-S was evidentially higher than that in the SDF-F (2.66 g/100 g versus 0.34 g/100 g), which suggested the release of pectic substances during steaming. The thermal activation of pectin esterase might be the reason for this [41], having been reported in the cases of the culinary cooking of carrots and onions [42,43].

#### 3.3.2. FT-IR Spectra

The FT-IR spectra of the SDF-F and SDF-S are presented in Figure 4. The strong bands at around 3385 cm^−1^ were assigned to the O-H stretching vibration, indicating the existence of hydrogen bonds within or between polysaccharides. Weak bands around 2930 cm^−1^ were due to the asymmetric stretching vibration of C-H, which were typical absorption peaks of polysaccharide [44,45]. Notably, the SDF-S showed a unique peak at 1741 cm^−1^, which indicated the C=O stretching of ester bonds [30] and revealed the existence of pectin [46]. There were two bands that indicated the free carboxylate groups in sweet potato SDF: an asymmetrical stretching band around 1630 cm^−1^ and a weaker symmetric stretching band around 1413 cm^−1^ [47]. The peaks near 1367 cm^−1^ indicated the presence of sulfate groups in these two SDF [48,49]. The peaks at 1104, 1077, and 1035 cm^−1^ indicated the existence of stretching vibrations of C-C and C-O, which were attributed to glycosidic linkage between sugar units [50]. The peaks at 840 cm^−1^ and 910 cm^−1^ indicated there were α-glycoside bonds and β-glycoside bonds in both sweet potato SDFs, respectively [51,52], which might refer to the α-1,2, α-1,6, β-1,2, and β-1,6 bonds in their resistant dextrin components [53]. From the FT-IR spectra, the type and position of the main absorption peaks of SDF-F and SDF-S did not change substantially upon steaming, but the different intensity of absorption peaks indicated that the physicochemical and functional properties of sweet potato SDF may have changed.

### 3.4. Proliferation Effects of Sweet Potato SDF on Lactobacillus spp.

#### 3.4.1. Growth Stimulation

In the present study, we assessed the effects of the SDF-S and SDF-F on *Lactobacillus* spp. proliferation as carbon sources in non-carbohydrate MRS broth. Additionally, the results were compared with inulin or glucose as the sole carbon source. OD_600_ and pH of the cultures were recorded to evaluate the growth of *L. plantarum*, *L. reuteri*, *L. acidophilus*, and *L. rhamnosus* (Figure 5 and Figure 6). It presented that all the four *Lactobacillus* spp. reached the stationary phase after 12 h of culturing. The data of OD_600_ and pH during stationary phase were compared between different carbon source by using Kruskal–Wallis one-way ANOVA according to their distribution (Table 2). As the control, the OD_600_ and pH in the non-carbohydrate culture were not included in the statistics, since their values were rather away from the other four groups.

The OD_600_ of the four *Lactobacillus* spp. cultures at stationary phases showed the coincident order among the four carbohydrate groups, which was SDF-F > SDF-S > Glucose > Inulin. For the *L. plantarum* and *L. rhamnosus*, significant differences in OD_600_ could only be detected between the SDF-F and inulin group (*p* < 0.05). While for the *L. reuteri* and *L. acidophilus*, OD_600_ values of the SDF-F group and SDF-S group were significantly higher than the inulin group (*p* < 0.05), and OD_600_ of the SDF-F group was significantly higher than the glucose group (*p* < 0.05).

As reported in Table 2, the culture pH at stationary phases was in the order of *Inulin* > SDF-S > SDF-F > Glucose for the *L. plantarum L. reuteri*, and *L. rhamnosus*. Significant differences could be determined between the inulin and SDF-F group, and between the glucose and inulin group (*p* < 0.05). In the *L. acidophilus* culture, the pH at stationary phage was significantly higher in the inulin group than in the glucose group (*p* < 0.05) but was almost equal in the SDF-F and SDF-S group. The pH of the stationary *L. acidophilus* culture in the two SDF groups was almost equal and was higher than the glucose group but lower than the inulin group without significant difference.

The proliferation of the *Lactobacillus* spp. was measured with OD_600_ of the culture, and acidogenicity was assessed with the culture pH in the present study, both of which are important indices of the benefits of *Lactobacillus* spp. [54,55]. They are determined mainly on the hereditary property of *Lactobacillus* spp., but also associated with the quantity and quality of probiotic in the culture [56]. The changes of OD_600_ and pH values during the 24 h of culturing demonstrated that both SDF-S and SDF-F promoted the proliferation and acid production of *Lactobacillus* spp., and their effects were superior to inulin, the commercial prebiotic. The carbon source utilization during the 24 h culturing is reported in Table 3. It was shown that glucose was the favorite carbon source for the four *Lactobacillus* spp., followed by the SDF-F, SDF-S, and inulin in turn, with significant differences among the four carbohydrate groups (*p* < 0.05). Although more glucose was utilized, the SDF-F presented the superior effect on the proliferation and acidogenicity of the four *Lactobacillus* spp. (*p* < 0.05). These results were in accordance with the rather shorter lag growth phase in the SDF-F group than the others (Figure 5a–c). The different lag growth phase among the four *Lactobacillus* spp. cultured with same carbon source might result from their different hereditary and physiological properties (Figure 5).

It has been documented that DF with a low Mw had superior proliferation effects because it took less time for probiotics to hydrolyze low Mw fiber into less complex monosaccharides and utilize them for proliferation [57,58,59]. Judged from the OD_600_ and pH of culture, and utilization efficiency during the 24 h of culturing, SDF-S was the inferior carbon source to SDF-F for the four *Lactobacillus* spp. fermentation, despite its a low Mw. The composition of SDF may account for the special results. Neutral carbohydrates (e.g., glucose, galactose, xylose, and fructose) in indigestible carbohydrate polymers may provide greater prebiotic potential than polyuronic acid, since *Lactobacillus* spp. produced few pectinases [60,61,62]. The higher uronic acid concentration in SDF-S than in SDF-F might account for its lower proliferation effects on *Lactobacillus* spp.

#### 3.4.2. SCFA Profile

After 24 h of fermentation, SCFA in these *Lactobacillus* spp. cultures were analyzed. Additionally, the results are reported in Figure 7. Acetic acid, propionic acid, and butyric acid were the main SCFA produced by *Lactobacillus* spp. fermentation with these four carbon sources, of which acetic acid accounted for the highest proportion (Figure 7e).

As the favorite carbon source in this study, the glucose seemed to favor the acetic acid fermentation. As shown in Figure 7a, the acetic acid concentration in the four *Lactobacillus* spp. cultures was the highest in the glucose group. Additionally, the differences between the glucose group and the others were statistically significant (*p* < 0.05), except for those between the glucose and SDF-F group in the *L. rhamnosus* culture. The inulin group showed the lowest acetic acid fermentation capacity (Figure 7a).

SDF-F might be the better carbon source for propionic acid and butyric acid fermentation. As presented in Figure 7b, the propionic acid concentration in the *L. reuteri* culture was significantly higher in the SDF-F group than the others (*p* < 0.05) and was significantly higher than the inulin and glucose group in the *L. rhamnosus* culture (*p* < 0.05) and was also significantly higher than the inulin group in the *L. plantarum* culture (*p* < 0.05). The butyric acid concentration in the *L. acidophilus* and *L. rhamnosus* cultures was significantly higher in the SDF-F group than the inulin group (*p* < 0.05). In the *L. plantarum* culture, the butyric acid concentration in the SDF-F group was also higher than the SDF-S group (*p* < 0.05). Apart from the significant differences mentioned above, no statistical difference could be detected between the two SDF groups. Nevertheless, it is worth noting that the concentration of the propionic acid and butyric acid in the SDF-S group were lower than the SDF-F group in all of the four *Lactobacillus* spp. cultures, which might suggest that the propionic acid and butyric acid fermentation capacity of SDF-S was inferior to the SDF-F.

The proportion of butyric acid and propionic acid in total SCFA is another indicator used to assess the prebiotic potential of DF. It has been reported that butyric acid and propionic acid have more beneficial impacts on intestinal health. Butyric acid provides energy for colonocytes and increases the production of mucin [63]. Propionic acid is mainly used as energy via liver metabolism after absorption by the colon. It lowers fatty acid content in the liver and plasma and exerts immunosuppressive actions [64]. Both propionic acid and butyric acid induce the production of intestinal hormones to regulate satiety, reduce food intake, and prevent obesity [65]. They can also exert anticancer and anti-inflammatory effects on inflammatory bowel disease, lung cancer, and necrotizing enterocolitis as studies investigated [66,67,68]. By calculating the proportion of propionic acid and butyric acid in SCFA, we confirmed the superior *Lactobacillus* spp. fermentation property of the SDF-F. In the four *Lactobacillus* spp. cultures, the summary percentage of these two fatty acids accounted for 26~35% of the total SCFA in the SDF-F group (Figure 7e). Additionally, they were 23~34%, 23% to 27%, and 21~26% in the inulin, SDF-S, and glucose groups, respectively.

## 4. Conclusions

This study revealed the influence of steaming treatment on the SDF content in the pulp of sweet potato, as well as its pronounced changes to the composition and functional properties of SDF. The 20 min steaming treatment brought about the biggest increase in the SDF content. The SDF of both fresh and 20 min steamed sweet potato were composed of over 90% glucose, while the SDF from steamed sweet potato showed a significantly higher uronic acid level and much smaller molecular weight than that of fresh sweet potato, which indicated more degraded pectin and resistant dextrin in it. The SDF from steamed potato presented an inferior proliferation effect on four tested *Lactobacillus* spp. than that of fresh sweet potato, though its effect was superior to inulin. The propionic acid and butyric acid production from the *Lactobacillus* spp. fermentation with SDF from steamed sweet potato as carbon source was much lower than that of fermentation with SDF from fresh sweet potato, and also slightly lower than that of fermentation with inulin. It can be concluded that steaming treatment increased the SDF content, but obtained SDF with lowered functional properties, as far the probiotic activity is concerned.

## Figures and Tables

**Figure 1 foods-12-01620-f001:**
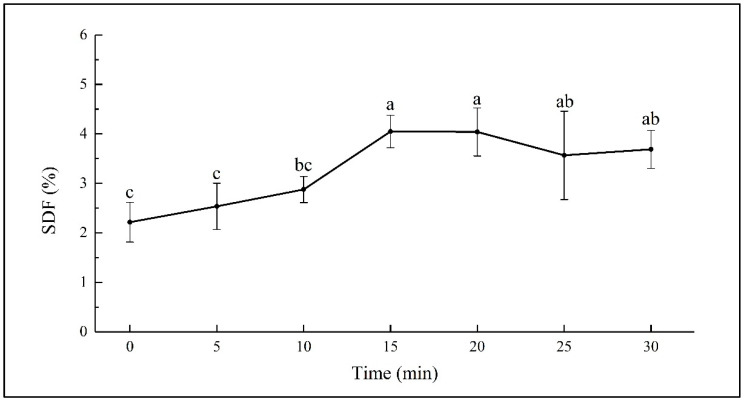
SDF content in sweet potato steamed for different times. Different letters on the error bars indicate significant differences (*p* < 0.05).

**Figure 2 foods-12-01620-f002:**
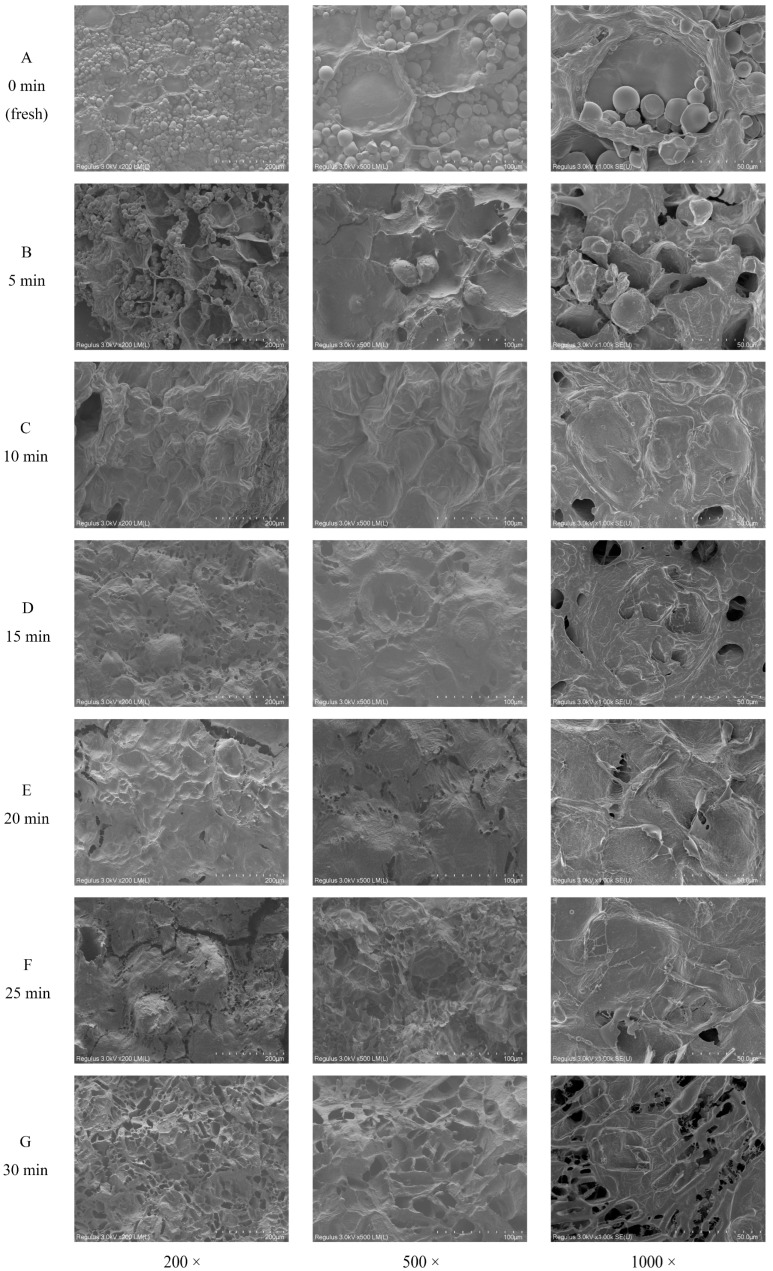
Morphologic observation of sweet potatoes during steaming. Scanning electron microscope images of sweet potato samples steamed for 0, 5, 10, 15, 20, 25, and 30 min are labeled with (**A**–**G**), respectively.

**Figure 3 foods-12-01620-f003:**
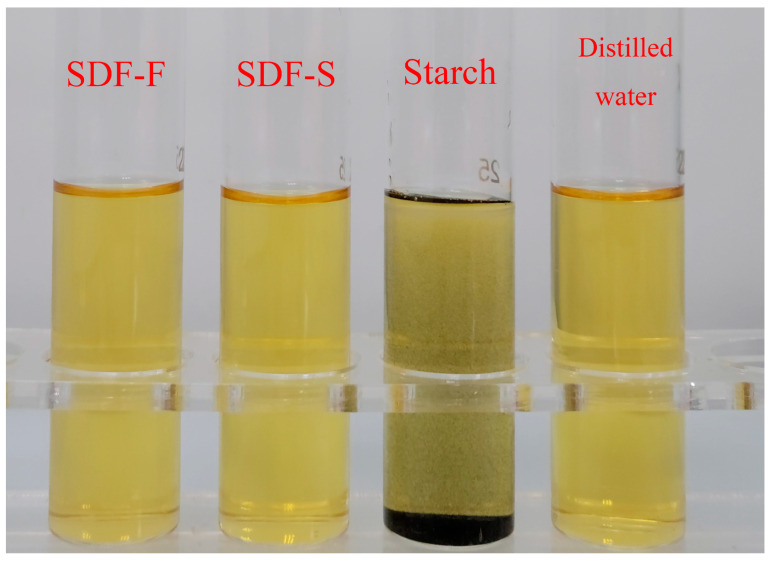
Iodine reaction of sweet potato SDF solutions.

**Figure 4 foods-12-01620-f004:**
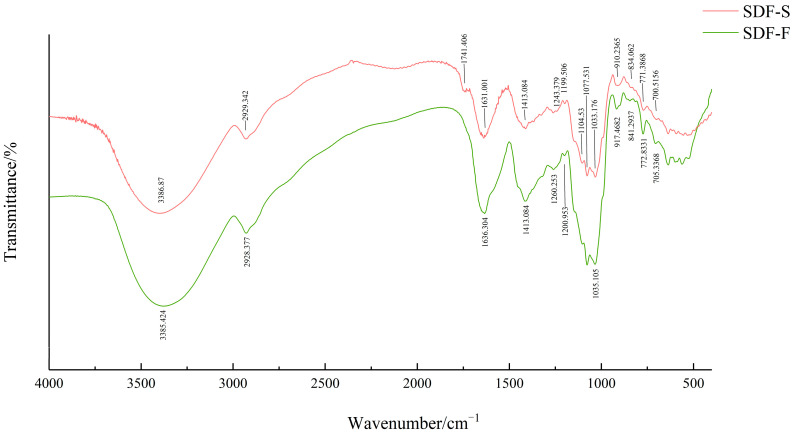
FT-IR spectra of sweet potato SDF.

**Figure 5 foods-12-01620-f005:**
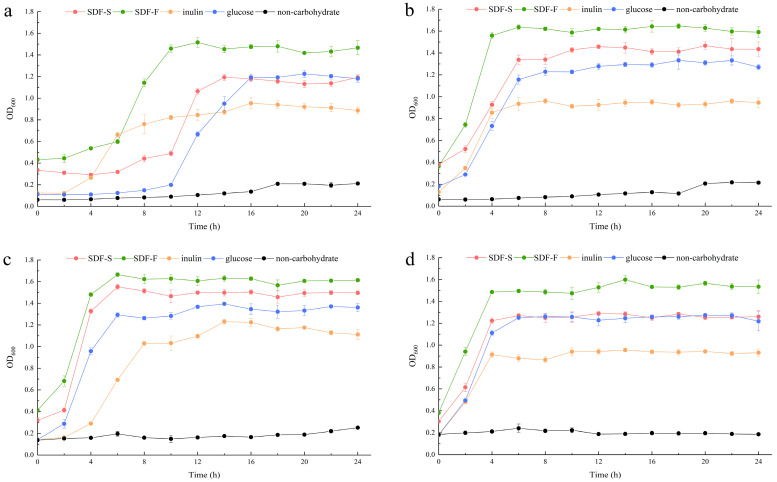
Growth curves of the *Lactobacillus* spp. with different carbon sources. (**a**) *L. plantarum*; (**b**) *L. reuteri*; (**c**) *L. acidophilus*; (**d**) *L. rhamnosus*.

**Figure 6 foods-12-01620-f006:**
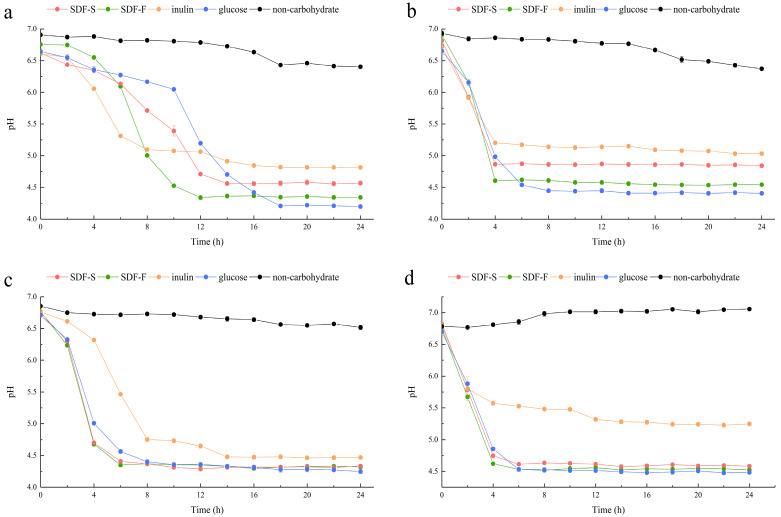
Changes in pH of the *Lactobacillus* spp. cultures with different carbon sources. (**a**) *L. plantarum*; (**b**) *L. reuteri*; (**c**) *L. acidophilus*; (**d**) *L. rhamnosus*.

**Figure 7 foods-12-01620-f007:**
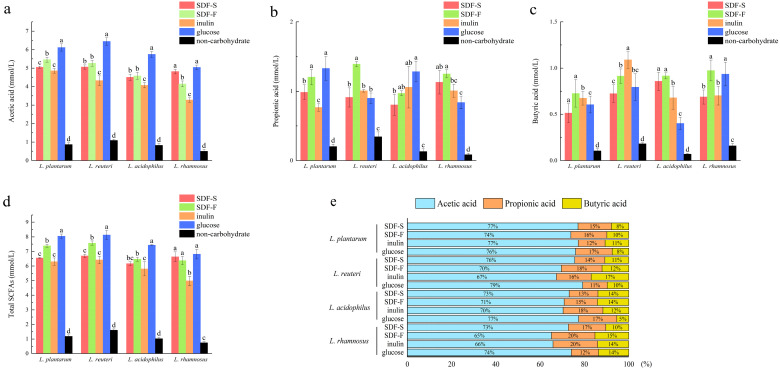
SCFA profile of four *Lactobacillus* spp. in culture media with different carbon sources after 24 h of fermentation: (**a**) the acetic acid concentration in culture media of four *Lactobacillus* spp. with different carbon sources, (**b**) the propionic acid concentration in culture media of four *Lactobacillus* spp. with different carbon sources, (**c**) the butyric acid concentration in culture media of four *Lactobacillus* spp. with different carbon sources, (**d**) the total SCFA concentration in culture media of four *Lactobacillus* spp. with different carbon sources, (**e**) the proportion of acetic acid, propionic acid, and butyric acid in total SCFA. For a certain *Lactobacillus* spp. fermentation, different letters on the columns represent significant differences in the SCFA concentration among four carbon source groups (*p* < 0.05).

**Table 1 foods-12-01620-t001:** Chemical composition of sweet potato SDF.

Item	SDF-S	SDF-F
Neutral carbohydrate (g/100 g)	59.31 ± 0.73 ^a^	46.83 ± 0.63 ^b^
Uronic acid (g/100 g)	25.36 ± 0.48 ^a^	9.60 ± 0.14 ^b^
Total carbohydrate (g/100 g)	84.68 ± 0.64 ^a^	56.43 ± 0.76 ^b^
Iodine reactions	Negative	Negative
Molecular weight (kDa)	5.32	28.79
Mw/Mn	1.61	1.35
Monosaccharide composition (g/100 g)		
Glc	91.88	95.73
Gal	3.14	1.77
Ara	0.89	0.53
Rha	0.65	0.36
Man	0.42	0.89
Rib	0.09	0.12
Fuc	0.12	not detected
GlcA	0.14	0.26
GalA	2.66	0.34

Note: The data are expressed as mean ± SD. The data with different letters within the same row are significantly different at *p* < 0.05. Glc, glucose; Gal, galactose; Ara, arabinose; Rha, rhamnose; Man, mannose; Rib, ribose; Fuc, fucose; GlcA, glucuronic acid; GalA, galacturonic acid.

**Table 2 foods-12-01620-t002:** Average OD_600_ and pH at stationary phases of four *Lactobacillus* spp. with different carbon sources.

	OD_600_				pH			
	SDF-S	SDF-F	Inulin	Glucose	SDF-S	SDF-F	Inulin	Glucose
*L. plantarum*	1.17 ± 0.03 ^ab^	1.46 ± 0.03 ^a^	0.92 ± 0.03 ^b^	1.19 ± 0.10 ^ab^	4.57 ± 0.08 ^ab^	4.35 ± 0.01 ^b^	4.82 ± 0.04 ^a^	4.22 ± 0.20 ^b^
*L. reuteri*	1.44 ± 0.02 ^ab^	1.62 ± 0.02 ^a^	0.95 ± 0.01 ^c^	1.30 ± 0.02 ^bc^	4.86 ± 0.01 ^ab^	4.54 ± 0.01 ^bc^	5.08 ± 0.04 ^a^	4.41 ± 0.01 ^c^
*L. acidophilus*	1.50 ± 0.02 ^ab^	1.61 ± 0.02 ^a^	1.17 ± 0.05 ^c^	1.35 ± 0.03 ^bc^	4.32 ± 0.01 ^ab^	4.32 ± 0.01 ^ab^	4.47 ± 0.01 ^a^	4.28 ± 0.03 ^b^
*L. rhamnosus*	1.26 ± 0.02 ^ab^	1.54 ± 0.03 ^a^	0.94 ± 0.01 ^b^	1.26 ± 0.02 ^ab^	4.59 ± 0.01 ^ab^	4.54 ± 0.01 ^bc^	5.24 ± 0.02 ^a^	4.49 ± 0.01 ^c^

Note: The data are expressed as medium ± SD. Significant differences among groups was determined by using Kruskal–Wallis test. The data with different letters within the same row are significantly different at *p* < 0.05.

**Table 3 foods-12-01620-t003:** Carbon source utilization ratio of four *Lactobacillus* spp. after 24 h of fermentation.

Carbohydrate Source	Utilization Ratio (%)			
	*L. plantarum*	*L. reuteri*	*L. acidophilus*	*L. rhamnosus*
SDF-S	83.41 ± 0.15 ^c^	86.83 ± 0.14 ^c^	89.09 ± 0.52 ^c^	92.11 ± 0.50 ^c^
SDF-F	90.08 ± 1.27 ^b^	93.50 ± 0.63 ^b^	90.92 ± 0.76 ^b^	95.16 ± 0.60 ^b^
Glucose	96.03 ± 0.52 ^a^	97.22 ± 0.16 ^a^	94.43 ± 0.48 ^a^	96.89 ± 0.48 ^a^
Inulin	47.03 ± 0.77 ^d^	58.91 ± 2.34 ^d^	79.59 ± 0.76 ^d^	55.50 ± 0.65 ^d^

Note: The data are expressed as mean ± SD. The data with different letters within the same column are significantly different at *p* < 0.05.

## Data Availability

Data is contained within the article.

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
