# Peer review of "Effects of Steaming on Sweet Potato Soluble Dietary Fiber: Content, Structure, and Lactobacillus Proliferation In Vitro"

_foods, 2023, doi:10.3390/foods12081620_

Round 1

Reviewer 1 Report

Line 112-113. SDF was prepared, using ultrasonic-assisted extraction, and the parameters were selected according to our previous study. Could you please put the reference where to consult these parameters, or has it not been published? If so, please indicate it with data not shown.

Linea 116 bath processor at 400 W and 40 °C for 40 min. Given the conditions, surely the temperature in your experiment had changed. How did you control the temperature changes to maintain 40 C for 40 min?

Line 116 bath processor.  Please specify the brand, model, city, and country of origin.

Line 119-121. Significant changes were observed in SDF content during steaming processing (p < 0.05). As shown in Figure 1, SDF content increased significantly from 2.21% to 4.04% during the first 15-min steaming (p < 0.05).Is there a significant change? According to the way in which you represent a statistical analysis, it would seem that at least the first 10 min, there is no change since they share the same literal (b). Perhaps this part of the discussion should be correctly modified to HIGHLIGHT the SDF increment.

Please improve table 2.

Please improve figures from 5 to 8. Although they explain your results, they are difficult to read with the naked eye.

Reviewer 2 Report

General overview.

Manuscript foods- titled “Effects of steaming on sweet potato soluble dietary fiber: con-1 tent, structure, and Lactobacillus proliferation in vitro”, investigate the influence of steaming on sweet patato dietary fiber. The manuscript is very interesting considering the different analytical approaches applied. However, some modifications should be made in particular at regard to the statistical elaboration.

Major comment

Materials and methods

Considering your dataset parametric ANOVA is not the best statistical test to check the differences. Please run the variances analyses with non parametric test such as Kruskal-Wallis.

Result and discussion

All data measure unit should be reported in SI. Please replace % as g / 100g. In addition all data should be reported on Dry Weight to compare the different matrix excluding the water dilution effects.

Minor

Keyword: I suggest to use different word respect to the title

Introduction:

Line 24-31 it would be appreciated to distinguish the chemical composition of the peel from that of the pulp

Line 32 replace resistant with indigestible polysaccharides

Round 2

Reviewer 1 Report

The authors have made the suggested changes to the document.

Reviewer 2 Report

No additional comment